



# An errors-in-variables extreme-value model for estimating interpolated extreme streamflows at ungauged river sections

Duy Anh Alexandre[1,2] and Jonathan Jalbert[1]

[1]Department of Mathematics and Industrial Engineering, Polytechnique Montréal
[2]Geosapiens

**Correspondence:** Jonathan Jalbert (jonathan.jalbert@polymtl.ca)

**Abstract.** Estimating extreme streamflows is critical for delimiting flood zones and designing fluvial infrastructure, but for the vast majority of river sections, no measurements are available. Estimated streamflows at ungauged river sections using spatial interpolation and hydrological modeling are uncertain, and in the context of extreme value analysis, this uncertainty can be crucial when estimating return levels. In the present paper, an errors-in-variables extreme value model is proposed to account for the estimated streamflow uncertainty at ungauged river sections. The true unobserved streamflows correspond to the missing variables in a Bayesian hierarchical model. In this model, the uncertainty of the unobserved streamflows propagates to the uncertainty in the estimated return levels. The model was implemented to estimate the streamflow return levels of 211 ungauged sections of the Chaudière River watershed in Southern Quebec, Canada.

## 1 Introduction

In hydrology, estimating extreme streamflows is essential for sustainable and optimal water management. Extreme streamflow estimations are notably used for providing dimensions of infrastructures like dams and for the delimitation of flood zones. Flooding is a phenomenon that is expected to be increasingly exacerbated by the effects of climate change (*e.g.* Asadieh and Krakauer, 2017).

The extreme value theory (see for example Coles, 2001) is a statistical framework used to estimate extreme streamflows from observations. It provides a set of statistical models that allow for rigorous estimation of the extreme values corresponding to return periods exceeding the range of observations. A classic extreme value model is the block maxima model, where, in environmental applications, maxima are typically taken over a one-year period. Therefore, let $Y_1, \ldots, Y_n$ denote a series of $n$ streamflow annual maxima at a river section. The Fisher–Tippett–Gnedenko theorem (Fisher and Tippett, 1928; Gnedenko, 1943) suggests that, if the block size is sufficiently large, the distribution of $Y_j$ for $1 \le j \le n$ can be approximated by the generalized extreme value (GEV) distribution as





follows:

$$\mathbb{P}(Y_j \leq y) \approx \exp\left[-\left\{1+\xi\left(\frac{y-\mu}{\sigma}\right)\right\}_+^{-1/\xi}\right] \tag{1}$$

where $a+ = \max\{0,a\}$ and $\mu \in \mathbb{R}$, $\sigma > 0$, and $\xi \in \mathbb{R}$ denote respectively the location, scale, and shape parameters

of the GEV distribution. The shape parameter $\xi$ is of particular interest for extrapolation because it determines the behavior of the right tail of the underlying distribution. A positive value suggests a heavy tail, a null value corresponds to an exponentially decreasing tail, while a negative value corresponds to a right-bounded distribution.

In environmental applications, a commonly used risk measure is the $T$-year return level $r_T$, where $T > 0$ is the return period. This measure is defined as the value that is expected to be exceeded once every $T$ years, assum-

ing climate stationarity. The $T$-year return level thus corresponds to the quantile $G(r_T \mid \mu,\sigma,\xi) = 1-1/T$ of the GEV distribution, where $G(\cdot \mid \mu,\sigma,\xi)$ denotes the GEV cumulative distribution function with parameters $(\mu,\sigma,\xi)$ as expressed in Eq. 1. By inverting the cumulative distribution function, the expression for $r_T$ is as follows:

$$r_T = \mu - \frac{\sigma}{\xi}\left[1-\left\{-\log\left(1-\frac{1}{T}\right)\right\}^{-\xi}\right]. \tag{2}$$

To account for the potential non-stationarity of the annual discharge maxima induced by climate change, the

parameters of the GEV distribution can vary according to various explanatory variables (e.g. Coles, 2001, Chap. 5). In this case, the $T$-year *effective return levels*, which correspond to the set of quantiles of order $(1-1/T)$ for each of these years, can be computed (Katz et al., 2002). The effective $T$-year return level of year $i$ can be interpreted as the $T$-year return level if the climate of year $i$ were to continue indefinitely.

This extreme value model cannot be directly implemented to study extreme streamflows in ungauged river sections.

As streamflow measurement is particularly difficult, the vast majority of river sections are ungauged. Estimating streamflow in ungauged rivers has thus been an important area of research in hydrology for many decades (e.g. Razavi and Coulibaly, 2013). In Canada, streamflow estimation at ungauged rivers is challenging due to sparse hydrometric and meteorological networks (Spence et al., 2013). Traditionally, two families of approaches have been defined based on whether they use hydrological models for estimating streamflows at ungauged locations or not.

Hydrological model-based methods involve a calibration step of model parameters using recorded data from selected gauged river sections. These adjusted model parameters are then applied at ungauged sites to estimate streamflows (Li et al., 2015). Methods that do not use hydrological models primarily rely on statistical approaches to interpolate the characteristics of observations (e.g., T-year return level) to ungauged sites with similar hydrological character-istics. These methods are generally not suitable for estimating daily streamflow series. Recently, Lachance-Cloutier

et al. (2017) developed a statistical method for streamflow interpolation at ungauged river sections using several hydrological model simulations as covariates. The method combines both families of approaches by using outputs from hydrological models and nearby observed streamflows to statistically interpolate streamflow series at ungauged



sites. It was shown that the method developed by Lachance-Cloutier et al. (2017) outperforms other conventional approaches (such as nearest neighbor, direct use of hydrological model outputs, ordinary kriging, and topological kriging), particularly when the ungauged site is relatively distant from the nearest gauged sites.

A classical frequency analysis, as described in the previous paragraphs, is not suitable for extrapolating interpolated streamflows to long return periods at ungauged sites. This is because it would ignore substantial interpolation uncertainty. It is desirable to account for interpolation uncertainty to properly reflect the true confidence level of the estimates of interest. Assuming that input data used for statistical inference are exact when they are actually accompanied by errors is misleading and can lead to overconfidence in the precision of model estimates. Therefore, uncertain data pose an additional challenge for flood frequency analysis, which already faces estimation uncertainty due to the scarcity of information inherent in extreme values.

Traditionally, probabilistic approaches to account for uncertainty in input streamflow data in extreme value analysis are quite scarce. Uncertainty analysis in climatology and hydrology typically involves sensitivity analysis, scenario analysis, or Monte Carlo simulation analysis, but lacks a systematic modeling of all information (W. Katz, 2002). For instance, in sensitivity analysis, the same model is applied to different versions of input data independently, and then the estimated quantities are compared and analyzed to assess their variability. This is usually conducted using analysis of variance techniques, which calculate the impact of each independent source of uncertainty on the variability of the estimated quantities (e.g. Yip et al., 2011; Giuntoli et al., 2018). Meanwhile, W. Katz (2002) also argued that "anything less than a fully probabilistic approach to uncertainty analysis is inadequate" and advocates for Bayesian statistical methods to account for uncertainty in environmental data.

A common approach to probabilistically model data uncertainty, dating back to Kalman (1960), is to assume that imperfect observations are related to an unobserved hidden variable that represents the truth. This unobserved variable typically forms the latent layer of a hierarchical model. In regression analysis, errors-in-variables models (e.g. Casella and Berger, 1990) are commonly used based on this principle. In such models, each observation is decomposed into an unknown true value (latent or missing variable) and an observation error. The total error thus originates from two sources: a modeling error term and an observation error term.

Regression errors-in-variables approaches have been applied in climate change detection and attribution to accommodate uncertainties in climate modeling (e.g. Huntingford et al., 2006; Ribes et al., 2017). Recently, in the context of developing scoring rules for climate forecast evaluation, Bessac and Naveau (2021) proposed a new scoring rule scheme that incorporates errors in the observed verification data through a hidden variable. The final metric used is the conditional mean of scores conditioned on the observation errors.

Incorporation of data uncertainty into the extreme value analysis framework is less developed. Data uncertainty associated with measurement errors has received particular attention in the literature, especially when paleohydrologic or historical flood data is used in extreme value analysis (Kuczera, 1996; O'Connell et al., 2002; Petersen-Øverleir and Reitan, 2009; Reis and Stedinger, 2005). Kuczera (1996) proposes a modification to the likelihood function to model errors involved in the calculation of rating curves. This necessity arises because historical data typically





exhibits non-negligible errors (high uncertainty, truncated data). Neppel et al. (2010) proposed a flood frequency analysis model for recent and historical flood data, considering that discharge values are estimated using a hydraulic model and are subject to rating curve uncertainty. The error models used by these authors typically assume that observed extreme flood data equals the true extremes multiplied by a log-normally distributed random variable with unit mean and known variance.

The objective of this paper is to develop an extreme value model for interpolated streamflows at ungauged sites that accounts for interpolation uncertainty. This model will be used to estimate T-year return levels for more than 211 river sections in a watershed located in southern Quebec. A set of interpolated streamflows from six different configurations of a hydrological model obtained from Lachance-Cloutier et al. (2017) will be modeled. The resulting frequency analysis takes into account the interpolation uncertainty of each member of the ensemble by combining the information provided by the different hydrological configurations. The model is fully probabilistic and leverages the Bayesian framework to model uncertainty in the input data. All parameters are simultaneously estimated using MCMC simulation methods. Bayesian inference allows for easy incorporation of prior knowledge, as well as full propagation of uncertainty from the interpolated streamflows to the final estimates of GEV parameters and streamflow return levels. This is crucial for extreme data, since the shape parameter is particularly sensitive to data uncertainty (Coles and Powell, 1996). Misestimation of the shape parameter can lead to false conclusions about the nature of extreme streamflow behaviors (Coles, 2001; Coles and Tawn, 1996). The model was initially developed to meet the operational need of analyzing the streamflow dataset interpolated by the method from Lachance-Cloutier et al. (2017). However, it can be utilized in a much broader context whenever an extreme value analysis is needed for uncertain data. In fact, the model can accept any collection of probabilistic error distributions as input, representing data uncertainty.

The remainder of this paper is organized as follows. Section 2 describes the data. Section 3 explains the statistical model. Section 4 presents results using data from the Chaudière River watershed situated in southern Quebec, which is used as a case study. Section 5 discusses the advantages and shortfalls of our methodology and finally the conclusion is provided. The code and data used are available for open access on the following repository: https://github.com/jojal5/Publications.

## 2 Data

The proposed methodology is applied to the 211 ungauged river sections of the Chaudière River watershed with a drainage area above 50 square kilometers. The Chaudière River watershed, shown in Figure 1, is located in southern Quebec and covers 6713 km$^2$, stretching from the American border in the south to Quebec City in the north. The watershed includes 78 municipalities and 179,000 inhabitants.

The estimated daily streamflows for these river sections were provided by the Ministère de l'Environnement, de la Lutte contre les changements climatiques, de la Faune et des Parcs (MELCCFP) from the Quebec provincial



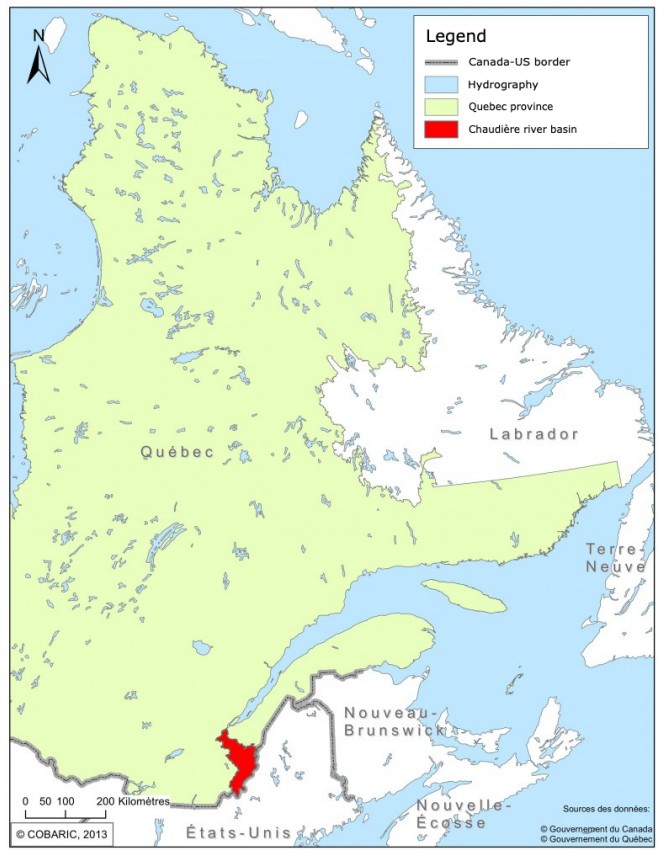

**Figure 1.** Chaudière river basin location. Source: the Chaudière river basin Committee (COBARIC) website.

government. These streamflows were estimated using hydrological simulations and data assimilation, as described in the following sections.

## 2.1 Simulated streamflows

At the ungauged river sections, streamflow estimates were obtained using the Hydrotel hydrological model (Fortin
et al., 2001). Hydrotel is a semi-distributed model that deterministically estimates daily streamflows using observed temperatures, precipitation, and geo-referenced data such as altitude, land use, soil type, and the network of rivers and lakes.

To account for Hydrotel calibration uncertainty, six different configurations of the Hydrotel model were used to consider parameterization uncertainty in hydrological response to meteorological forcings, generating six series of
interpolated streamflows from 1961 to 2020 at each river section. These six series of daily streamflows are referred to with the following acronyms: LN24HA, MG24HA, MG24HI, MG24HK, MG24HQ, and MG24HS. These configurations differ in the parameterization of the main hydrological processes. Two configurations were obtained through





global automatic calibration, while the other four were adjusted manually with hydrological expertise. Simulated streamflows were compared to historical observed daily values from 1961 to 2020 over more than 80 measuring
stations. Using the Kling-Gupta efficiency scores (Gupta et al., 2009) for validation, performances were found to be similar across the six configurations (Direction de l'hydrologie et de l'hydraulique, 2024).

## 2.2   Data assimilation

Since there is a discrepancy with the *in situ* observed streamflows, a data assimilation technique was used to correct the simulated streamflows (Lachance-Cloutier et al., 2017). The spatial correlation between the observations and
the estimated streamflows at gauged sites is used to correct the streamflows at ungauged sites. This procedure is repeated independently for the six hydrological model configurations.

   Data-assimilated streamflows at ungauged sites are uncertain. This uncertainty is quantified using the Gaussian distribution for the log daily streamflows for each hydrological model configuration (Lachance-Cloutier et al., 2017). Therefore, the data provided by the MELCCFP consists of six series of Gaussian parameters modeling the daily
streamflows for every ungauged river section from 1961 to 2020.

## 2.3   Annual maxima distribution of streamflows and notation

At a given ungauged river section, let $X_{ij}$ denote the streamflow annual maximum for year index $1 \le j \le n$ and hydrologic model configuration index $1 \le i \le 6$. From the provided data, we do not have $X_{ij}$ directly, but we do have the associated Gaussian distribution parameters $(\eta_{ij}, \zeta_{ij})$ for the log streamflow:

$$\log X_{ij} \sim \mathcal{N}(\eta_{ij}, \zeta_{ij}). \tag{3}$$

Therefore, the streamflow distribution corresponds to the log-normal distribution:

$$X_{ij} \sim \mathcal{L}og\mathcal{N}(\eta_{ij}, \zeta_{ij}). \tag{4}$$

Figure 2 displays the estimated mean streamflows along with the interquartile intervals at the watershed outlet for (a) the first hydrological configuration and (b) all configurations. One can see that the uncertainty is quite large and
must be taken into account in the extreme value analysis.

## 3   Methodology

The goal of this paper is to perform an extreme value analysis of the series of six estimated annual streamflow maxima for each river section. We propose an original errors-in-variables extreme value model tailored to account for the uncertainty of the estimated streamflows in the extreme value analysis. The model also combines information
from the six hydrologic model configurations. The following sections describe the statistical model.



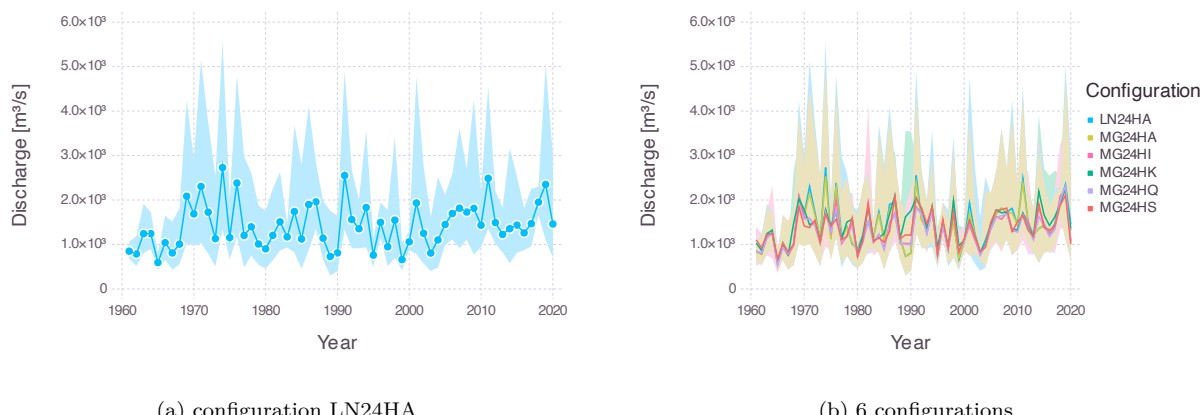

            (a) configuration LN24HA                            (b) 6 configurations

**Figure 2.** Annual maxima distributions extracted from the ungauged streamflow data at the Chaudière watershed outlet for (a) a single hydrologic configuration and (b) the six configurations ensemble. The ribbons correspond to the interquartile intervals of the Log-Normal distributions.

### 3.1 Missing value

Let $Y_i$ denote the true streamflow annual maximum of year $1 \leq i \leq n$ at a given river section, and let $\boldsymbol{Y} = (Y_1, \ldots, Y_n)$ denote the vector of annual maxima at this same river section. If $\boldsymbol{Y}$ were observed, then the GEV distribution expressed in Eq. (1) could be fitted to the data:

$$f_{(Y_i|\mu,\sigma,\xi)}(y_i) = \mathcal{G}EV(y_i \mid \mu, \sigma, \xi), \tag{5}$$

where $\mathcal{G}EV(y \mid \mu, \sigma, \xi)$ denotes the density of the GEV distribution with parameters $(\mu, \sigma, \xi)$ evaluated at $y$, and return levels could be estimated. However, these maxima are not observed but estimated. The variables $\boldsymbol{Y}$ thus correspond to missing values in the statistical model.

### 3.2 Model for the estimated streamflows

In the current framework, at a given river section, the data provided by the MELCCFP corresponds to a series of parameters $(\eta_{ij}, \zeta_{ij})$, where $\eta_{ij}$ and $\zeta_{ij}$ denote, respectively, the mean and standard deviation of the estimated log streamflow for year $1 \leq i \leq n$ and hydrologic configuration $1 \leq j \leq 6$. Assuming that the six hydrological configurations give independent interpolated discharge realizations as mentioned in Lachance-Cloutier et al. (2017), then the information coming from these configurations can be pooled together for the annual maximum discharge of year $j$ as follows:

$$f_{(y_i|,\boldsymbol{\eta}_i,\boldsymbol{\zeta}_i)}(y_i) \propto \prod_{j=1}^{6} \mathcal{L}og\mathcal{N}(y_i \mid \eta_{ij}, \zeta_{ij}), \tag{6}$$



where $\boldsymbol{\eta}_i = (\eta_{ij} : 1 \leq j \leq 6)$ and $\boldsymbol{\zeta}_i = (\zeta_{ij} : 1 \leq j \leq 6)$. In this model, each hydrological configuration contributes to the distribution of $Y_j$, but not equally. The configurations with small $\zeta_{ij}$ are more informative than those with large $\zeta_{ij}$. In other words, a hydrological configuration that provides a large uncertainty for the ungauged streamflow does not contribute as much as a configuration that provides less uncertainty.

As shown in Section 3.5, this model assumes that the 6 different hydrological model configurations are condionnally indenpendent on the true unobserved discharge. This is a commonly used assumption in hierarchical models (*e.g.* Gelman et al., 2013).

### 3.3 Annual maximum conditional distribution

Combining Eq. (5) and (6), the conditional distribution of the unobserved annual maximum $Y_j$ of year $j$ is obtained by pooling the information from the extreme value model and from the interpolated discharges distribution as follows:

$$f_{(Y_j | \boldsymbol{\eta}_j, \boldsymbol{\zeta}_j, \mu, \sigma, \xi)} \propto \prod_{i=1}^{S} \mathcal{L}og\mathcal{N}(y_j \mid \eta_{ij}, \zeta_{ij}) \times \mathcal{G}EV(y_j \mid \mu, \sigma, \xi). \tag{7}$$

The first term of this distribution combines the information provided by the different hydrological configurations. The second term allows the transfer of information from one year to another since the parameters of the GEV distribution are common for the maxima of all years.

### 3.4 Bayesian inference

The statistical model expressed in Eq. (7) contains the unknown GEV parameters $(\mu, \sigma, \xi)$ and the missing variables $\boldsymbol{Y}$ to be estimated. The Expectation-Maximization algorithm (Dempster et al., 1977) could have been implemented to estimate the parameters. Instead, we chose to perform Bayesian inference to tackle the missing value model since it straightforwardly provides the uncertainty on the missing values, the parameters, and all related quantities such as the return levels.

To perform Bayesian inference, a prior distribution for the model parameters is needed. We propose to use the following semi-informative prior for the GEV parameters:

$$f_{(\mu, \sigma, \xi)}(\mu, \sigma, \xi) \propto \sigma^{-1} \mathcal{B}eta\left(\xi + \frac{1}{2} \middle| 6, 9\right), \tag{8}$$

where $\mathcal{B}eta(y \mid \alpha, \beta)$ denotes the density of the Beta distribution of parameters $(\alpha, \beta)$ evaluated at $y$. This improper prior is non-informative for the location $\mu$ and scale $\sigma$ parameters of the GEV, but it is informative for the shape $\xi$ parameter. The Beta prior is inspired by Martins and Stedinger (2000), where the authors show that using this informative prior within a realistic range for the shape parameter improves the precision of its estimate for small sample sizes in environmental contexts. Sixty years of data is sometimes sufficient to estimate the GEV parameters



well, but in our case, the effective sample size is smaller since the annual maxima are uncertain. For these reasons, we chose to impose this informative prior distribution on the GEV shape parameter to improve the precision of the GEV parameter estimates. This improper prior for the GEV parameters yields a proper posterior, as shown by Northrop and Attalides (2016).

A sample from the posterior distribution of the missing values $Y$ and the GEV parameters $(\mu, \sigma, \xi)$ knowing $\boldsymbol{\eta} = (\boldsymbol{\eta}_i, \ 1 \le i \le n)$ and $\boldsymbol{\zeta} = (\boldsymbol{\zeta}_i, \ 1 \le i \le n)$, i.e. $f_{\{(\boldsymbol{Y}, \mu, \sigma, \xi)|\boldsymbol{\eta}, \boldsymbol{\zeta}\}}$ can be obtained by Markov Chain Monte Carlo. A Gibbs sampling scheme has been implemented.

### 3.5    Equivalent model representation

The proposed model can also be written in a more traditional fashion using a hierarchical model. Let

$$f_{(\eta_{ij}|Y_i = y_i, \zeta_{ij})}(\eta_{ij}) = \mathcal{N}\left(\eta_{ij} \,\middle|\, \log y_i, \, \zeta_{ij}^2\right)$$

denote the distribution of the parameter $\eta_{ij}$ for year $i$ and hydrological model configuration $j$, conditional on the true maximum $y_i$ and log streamflow variance $\zeta_{ij}$. Assuming independence of $\eta_{ij}$ across the 6 hydrological model configurations conditional on the true streamflow leads to the following joint distribution:

$$f_{(\boldsymbol{\eta}_i|Y_i = y = i, \zeta_{ij})}(\boldsymbol{\eta}_i) = \prod_{j=1}^{6} \mathcal{N}\left(\eta_{ij} \,\middle|\, \log y_i, \, \zeta_{ij}^2\right).$$

Using Eq. (5) for the distribution of $Y_i$, the joint distribution is as follows:

$$f_{\{(\boldsymbol{\eta}_i, Y_i)|\zeta_{ij}, \mu, \sigma, \xi\}}(\boldsymbol{\eta}_i) = \prod_{j=1}^{6} \mathcal{N}\left(\eta_{ij} \,\middle|\, \log y_i, \, \zeta_{ij}^2\right) \times \mathcal{G}EV(y_i \,|\, \mu, \sigma, \xi),$$

which is equivalent to Eq. (7).

Although this alternative model representation highlights the conditional independence assumption, it involves a somewhat cumbersome transformation from the log scale to the original streamflow scale. We believe that the

representation described in the previous sections is more intuitive for practitioners.

### 4    Results

Results are first presented in detail for the river section at the outlet of the Chaudière River watershed.

### 4.1    MCMC simulations

A sample of size 20,000 of the posterior distribution of the parameters and the missing variables (the series of annual

maxima) was generated with a Metropolis-within-Gibbs sampling scheme. The computation time was 3 seconds on





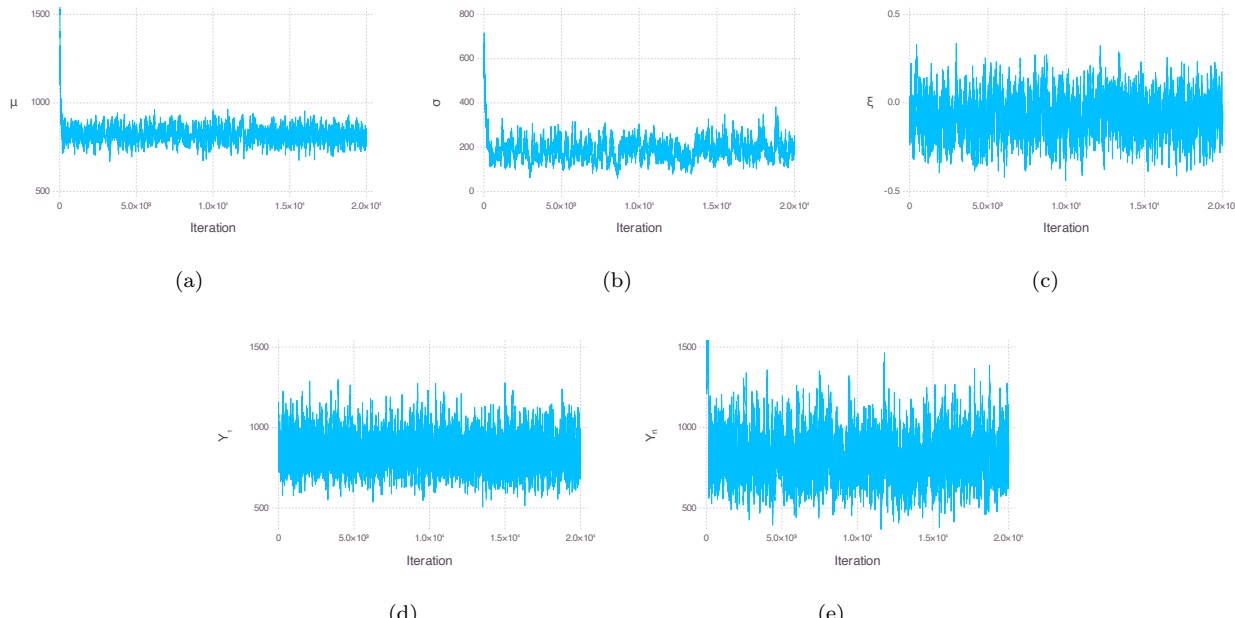

**Figure 3.** Traces of the GEV distribution parameters (a-c) and of two annual maxima of years 1961 (d) and 2020 (e), simulated using a Metropolis-within-Gibbs sampling scheme.

a personal computer with a 2.7 GHz Intel Core i7 processor. Figure 3 shows the traces of the GEV distribution parameters and of two annual maxima of years 1961 and 2020. These traces show the chain has well entered the sampling phase (reaching convergence) and that it exhibits good mixing. The first 10,000 iterations were discarded as the warm-up for the analysis.

## 4.2  Estimates of the annual maxima

Although this is not specifically the subject of the present article, which focuses more on frequency analysis with uncertain data, an interesting secondary result concerns the possible estimation of the unobserved annual maxima that constitute the missing variables in the model. In Figure 4, the posterior distributions of the annual maxima for the years 1961 and 2020 are represented by the solid red line, while the Log-Normal distributions obtained from the six hydrological configurations are represented by dotted blue lines. Using the proposed statistical model, the estimation of the maximum annual discharge utilizes information from all six hydrological configurations across all years, resulting in a decrease in uncertainty compared to the predictions from each individual configuration. Consequently, the estimations of the annual maxima are more precise. Note that since the posterior distribution of the annual maximum has a term proportional to the product of the six Log-Normal distributions, its mode remains close to the modes of the six Log-Normal distributions.





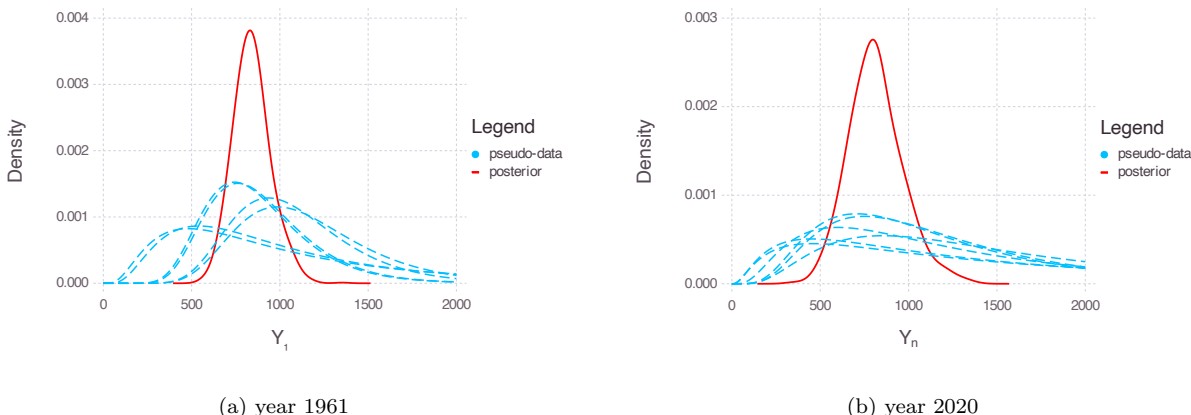

(a) year 1961                    (b) year 2020

**Figure 4.** The dashed lines illustrate the distributions of the annual maximum streamflows for the six configurations and the solid line shows the posterior distribution of the maximum streamflow for (a) year 1961 and (b) year 2020.

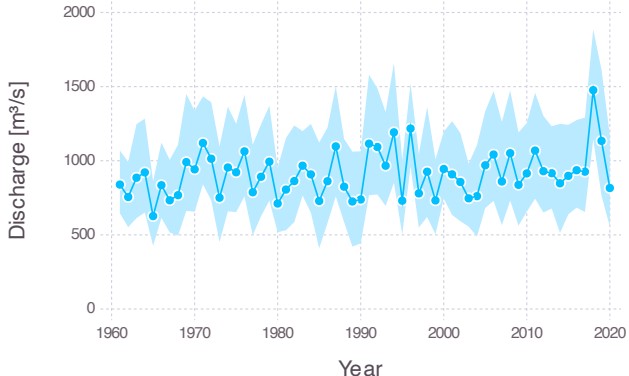

**Figure 5.** Posterior estimates of annual maxima at the outlet of the Chaudière river watershed and their 95% credible intervals.

Figure 5 shows the mean of the posterior distributions of the annual maximum discharge, along with the 95% credible intervals, at the outlet of the Chaudière watershed. Although the uncertainty in the annual maxima remains substantial, it has been reduced by combining information from the interpolated streamflows using the six hydrological configurations.





### 4.3 Estimates of the GEV parameters

Figure 6 shows the marginal posterior distributions of the GEV parameters. If the posterior mean is used as the point estimates, the location estimate is $\hat{\mu} \approx 820\,(744, 897)$, the scale estimate is $\hat{\sigma} \approx 179\,(106, 254)$ and the shape estimate is $\hat{\xi} \approx -0.08\,(-0.29, 0.15)$, where the values in parenthesis correspond to the marginal 95% credible intervals.

Figure 7 shows the diagnostic plots of the fitted GEV. These plots were obtained by standardizing the maxima sampled at each MCMC iteration to the Gumbel scale with the corresponding GEV parameters of these iterations. If the model fits well to the data, the residuals should be distributed according to the unit Gumbel distribution. As shown in Figure 7, the GEV distribution seems to fit very well to the estimated maxima.

### 4.4 100-year return level predictive distribution

Using the sample of the GEV parameters marginal posterior distributions generated by MCMC, a sample of the $T$-year return level predictive distribution can be computed. Figure 8 shows the predictive distribution of the 100-year return level at the outlet of the Chaudière watershed. The posterior mean is 1531 $m^3/s$ and the 95% credible interval is $(1238, 1973)$. This predictive distribution combines the uncertainty on the estimated maxima and the GEV parameters.

If the frequency analysis for the six configurations is performed separately, taking the median of the Log-Normal distributions as an estimate of the annual maxima, the average of the six return levels obtained would be 2429 $m^3/s$, which is much larger than the estimate obtained with the proposed model. The difference arises because the proposed model combines the information provided by the six hydrological configurations to perform the frequency analysis. Consequently, the uncertainty on the estimated annual maxima is considerably reduced compared to the very large uncertainties of the individual configurations.

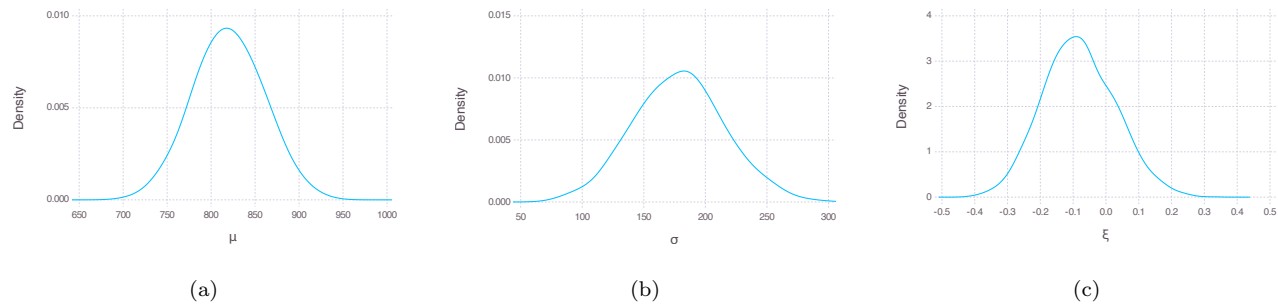

|       |       |       |
| (a) | (b) | (c) |

**Figure 6.** Marginal posterior distributions of the fitted GEV parameters, at the outlet of the Chaudière river watershed.





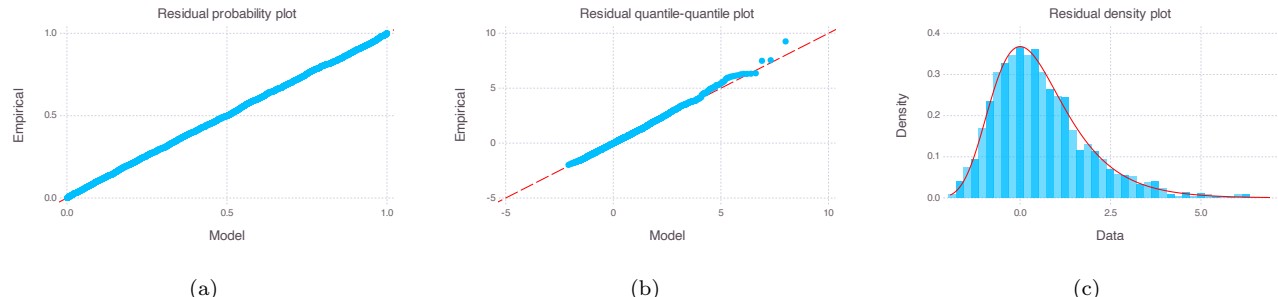

**Figure 7.** Residual diagnostic plots of the fitted GEV parameters, at the outlet of the Chaudière river watershed.

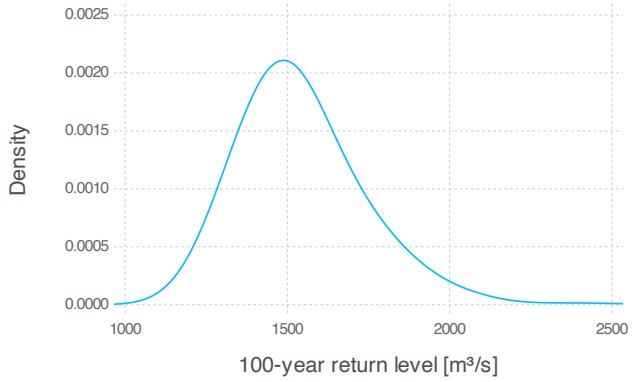

**Figure 8.** 100-year return level posterior distribution at the outlet of the Chaudière watershed.

## 270   4.5   Other river sections

Figure 9 shows the 100-year return level streamflows (a) and specific streamflows (b) for all sections of the Chaudière River. The specific streamflow is equal to the corresponding streamflow divided by the drainage area of the section, expressed in $m^3/s/km^2$. The Chaudière River flows from the Megantic Lake area (bottom corner) to the estuary of the Saint Lawrence River in Québec City (upper left corner). As expected, the flow intensity is lower for small tributaries and higher for the main branch of the Chaudière River, with an increasing gradient from upstream to downstream. The spatial pattern of 100-year return level specific streamflows is somewhat more complex. The increasing gradient from upstream to downstream is still present, but the highest specific streamflows are found in small sections of the Chaudière River, especially near the river's outlet (upper left corner). This could be explained by the smaller drainage areas of these sections compared to the main section.





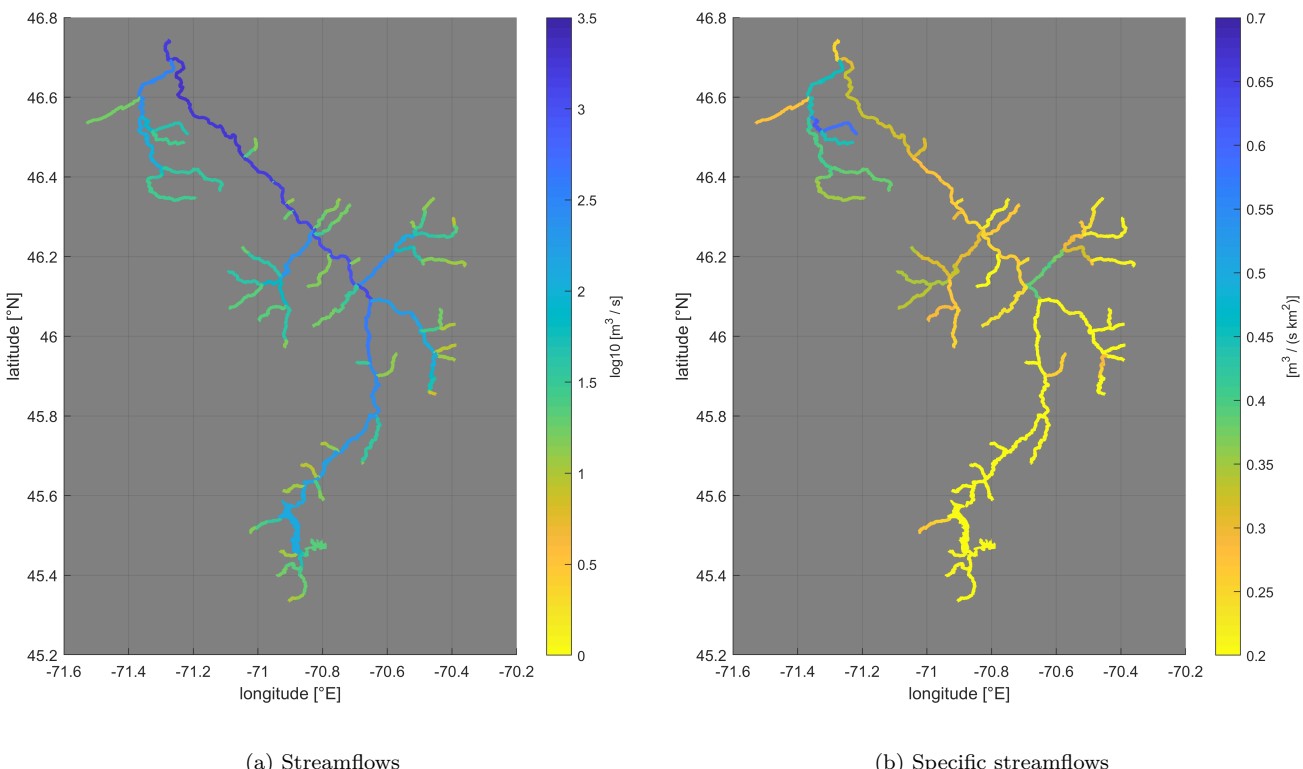

(a) Streamflows                                      (b) Specific streamflows

**Figure 9.** Estimated mean 100-year return level streamflows and specific streamflows for all sections of the Chaudière river. Values are shown on the logarithmic scale for streamflows.

## 5 Discussion

### 5.1 Non-stationary extensions

To account for potential non-stationarity in the annual maxima series, two non-stationary model extensions were considered: (1) the GEV location parameter as a linear function of the year; (2) the GEV location and log-scale parameters as a linear function of the year. According to the DIC, the best model, the one with the lowest DIC, is the stationary one. The computed DICs are respectively **-3140.9**, -3138.7, and -3139.3 for the stationary model and the non-stationary extensions.

### 5.2 Hydrological configurations independence assumption

The main assumption of the proposed model concerns the conditional independence assumption of the six hydrological configurations, each providing independent interpolated discharge realizations, conditional on the true unobserved streamflow. Hydrological arguments supporting this assumption are given by Lachance-Cloutier et al. (2017). This




assumption could be relaxed in several ways. One approach would be to modify the likelihood expressed in Eq. (6) by raising it to the power $0 < \alpha \leq 1$ in order to correct for the dependence between the hydrological configurations (see Sharkey and Winter, 2019, for an example in the context of a Bayesian hierarchical model for precipitation). However, estimating this additional exponent parameter would be very difficult when working with unobserved data, as it is the case in our framework. This is beyond the scope of the present paper.

Another way to relax the independence assumption would be to replace the density expressed in Eq. (6) by the following mixture of distributions:

$$f_{(y_j|,\boldsymbol{\eta}_j,\boldsymbol{\zeta}_j)}(y_j) = \frac{1}{S}\sum_{i=1}^{S} \mathcal{L}og\mathcal{N}(y_j \mid \eta_{ij}, \zeta_{ij}). \tag{9}$$

In this setting, no assumption on conditional independence is needed; it only assumes that each hydrological configuration is equally probable for modeling the annual maxima. This model has been implemented, but it appears that the mixture is not sufficiently informative for parameter estimation. Indeed, all the estimated annual maxima for each year tend to be equal to the GEV location parameter. Additionally, this model leads to serious identifiability issues during the MCMC procedure.

Therefore, the conditional independence assumption between hydrological configurations was retained. This choice is justified by hydrological arguments (Lachance-Cloutier et al., 2017) and is commonly used in hierarchical models, and attempting to relax this assumption would be challenging without introducing additional assumptions.

## 6 Conclusions

In this paper, we propose an errors-in-variables extreme-value model for conducting extreme value analysis on a set of interpolated streamflows obtained using different hydrological configurations in Southern Québec. The model addresses the uncertainty associated with estimating return levels, where the true, unobserved maxima serve as missing variables. It enables the estimation of annual maxima by integrating information from each hydrological configuration. Bayesian inference is employed to describe the posterior distribution of annual maxima series and GEV distribution parameters, with samples generated using MCMC simulation techniques.

The model successfully estimates return levels based on uncertain annual maxima, combining information from six hydrologic model configurations spanning 1961 to 2020. However, estimated annual maxima still exhibit considerable uncertainty due to interpolation errors in daily streamflow series. The posterior distribution of return levels incorporates this uncertainty into the analysis.Furthermore, the model's flexibility allows for accommodating various models of uncertainty in annual maxima, accommodating different sizes of interpolated streamflow datasets and offering multiple distribution options to model this uncertainty.

In future work, the conditional independence assumption of hydrologic model configurations could be relaxed using the approach notably employed by Sharkey and Winter (2019), which involves a magnitude correction to



the likelihood. Additionally, the spatial dependence of river sections within the same watershed could be modeled, particularly with a graphical model for extremes as proposed by Engelke and Hitz (2020). In the meantime, we believe that the developed approach meets the practical needs of engineers responsible for updating the mapping of
flood zones in Southern Québec.

*Code and data availability.*  The code and data to reproduce all the results and figures are available on the public repository https://github.com/jojal5/Publications.

*Author contributions.*  Authors' Contribution statement using CRediT with degree of contribution:

**Duy Anh Alexandre:** Formal Analysis (lead), Investigation (lead), Methodology (Lead), Software (equal), Validation
(equal), Visualization (equal), Writing – Original Draft Preparation (equal).

**Jonathan Jalbert:** Conceptualization (lead), Investigation (supporting), Funding Acquisition (lead), Methodology (supporting), Software (equal), Validation (equal), Visualization (equal), Supervision (lead), Writing – Original Draft Preparation (equal).

For more information, please see the taxonomy website.

*Competing interests.*  No competing interests are present.

*Acknowledgements.*  We would like to thank Jean-François Cyr, Simon Lachance-Cloutier, Édouard Mailhot and Charles Malenfant of the Direction de l'expertise hydrique of the Ministère de l'Environnement et de la Lutte contre les changements climatiques du Québec and David Huard and Gabriel Rondeau-Genesse of the Ouranos Consortium for the fruitful monthly meetings concerning this project. Their expertise was essential in the completion of this project. We would also like to thank
Gabriel Gobeil (Environment and Climate Change Canada) for his crucial support.

This work was supported by Natural Sciences and Engineering Research Council of Canada, Ouranos and INFO-Crue.





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
