# Peer review of "An errors-in-variables extreme-value model for estimating interpolated extreme streamflows at ungauged river sections"

_EGUsphere, 2024_

## Author Comment (AC1)

**EGUSPHERE-2024-2114**
**Detailed responses to Reviewer 1's comments**

Duy Anh Alexandre & Jonathan Jalbert

November 5, 2024

First, we would like to thank the reviewer who conducted a thorough review and provided relevant comments and suggestions. Due to its feedback and suggestions, we notably changed the model description in Section 3. The modifications in the revised version of the manuscript are highlighted in blue. Below are detailed responses to all of Reviewer 1's comments.

Reviewer 1 and Reviewer 2 suggest the use of observed discharges data to validate the results. However, it is not possible to validate the results as the reviewers suggest. The estimated discharges are obtained with a technique combining hydrologic models, interpolation and data assimilation. So, at the river sections where there is true observations, these observations are assimilated. So the real data cannot be used to validate in a strict senses the estimated discharges.

Estimated discharges at gauged sections still contain some uncertainty, primarily due to rating curve uncertainty when estimating discharges from water level measurements, especially for high discharge values. At ungauged sites, this uncertainty is greater and increases with the river distance from a gauging station. These sources of uncertainty affect the precision of return level estimates. In the manuscript, we aimed to show these impacts of uncertainty on return level estimations for both gauged and ungauged sections. To further emphasize this point in the revised version, we added Section 4.5 to analyze a gauged river section. This new analysis is twofold: first, it demonstrates the flexibility of the proposed approach to model uncertainty, and second, it shows the impact of including interpolation uncertainty in the return level estimates.

It would have been interesting to perform a cross-validation where real observations from a river section were withheld from assimilation, allowing us to compare the estimated return levels with the actual data and the estimated discharges. However, such cross-validation data is not available.

That being said, the estimated discharges were already evaluated at sites with observations in Lachance-Cloutier *et al.* (2017), but the emphasis was not on annual maxima.

**Summary**

The manuscript *An errors-in-variables extreme-value model for estimating interpolated extreme streamflows at ungauged river sections* by Duy Anh Alexandre and Jonathan Jalbert presents an interesting model with a slightly less traditional application in the realm of extreme value analysis. They focus on deriving information on extremes when these are not observed, employing information from hydrological models outputs and employing an errors-in-variables (EIV) approach.

The modeling approach is interesting and novel (as far as I know), but I do have concerns about the writing of both the text and the mathematical details, which I think make the manuscript not as clear as it should be and make me doubtful of the validity of the results.

Overall I think the manuscript presents and interesting approach but I would recommend the authors spend some time in re-reading the manuscript and in making sure the mathematical details

follow logically and are written clearly and with precision.

Thank you for your comment, which will help new readers understand our proposed approach. We thoroughly proofread the mathematical details to avoid typos and enhance clarity.

**Some more punctual comments:**

1. If I understand correctly there is no observed data used in any part of the model, but you use the simulated/estimated data (you use both terms, I think you mean the same thing) to derive the hypothetical/imputed annual maxima across the whole river network. Were the hydrological models not calibrated on something? Is there not any observed flow data that you can use to validate your modeling approach?

   It is correct that no observed data has been used in this model. Calibration and validation of the hydrological model's estimated discharges with observations were previously conducted by Lachance-Cloutier *et al.* (2017). The goal of this paper was to incorporate discharge uncertainty into the extreme value analysis. If no uncertainty is assumed at observed sites, a classical extreme value model can be used.

   We now use consistently *estimated discharges* in the revised version of the manuscript.

2. More importantly, you are modeling annual maxima: do you extract for each of the 6 model configurations a value for the year based on the maximum value of $\eta_{ij}$: do you have any way to ensure that you are modeling the flow value of the same day (and does it matter?)

   To extract the streamflow annual maxima for configuration $i$, we identify the maximum value $\max_{1 \leq k \leq 365} \eta_{ijk}$, where $\eta_{ijk}$ represents the location of the log-discharge distribution for configuration $i$, year $j$, and day $k$. The index $k$, which corresponds to the day on which the annual maxima occur, can differ for each configuration $i$ in a given year $j$, as no constraint was imposed. The rationale behind this approach is that annual maxima for a given year might not coincide across all hydrological configurations, though such short lags are not important for our purposes. We verified that the annual maxima were selected from the same period, with possible separation of only a few days. Our aim was to avoid retrieving a mix of Fall and Spring maxima within a given year at a river section.

   We added this clarification in the revised version of the manuscript.

3. In Section 3.2 you use $Y_i$ to indicate the maxima in year $i$, but then in 3.3 you use $Y_j$, with $j$ indexing the year, but you also have the eta and zeta parameters indexed by $i, j$, where now $i$ indicates the hyodrological model? This is quite confusing.

   You are right. This is an artifact from our working version of the manuscript. We now consistently use the index $i$ for the hydrological model configuration and $j$ for the year throughout the manuscript.

4. In Equation 6: should the logNormal not be for $X_{ij}$ - there is some unclarity in the notation here (the pendix of the $f$ should it not be big $Y$, we normally give distributions for r.v, not for realizations - indeed you do so in Eq. 7)?

   Thank you for pointing that out. The first $Y_j$ in Eq. 6 should indeed be capitalized. Following Point 8 of your review, we have decided to adopt a more traditional presentation of the model. We hope the distinction between random variables, realizations, and modeling assumptions is now clearer in Section 3.1 and 3.2 of the revised version of the manuscript.

5. Line 178 "contributes to the distribution of $Y_j$" $\rightarrow Y_i$? The idea is that in any year each jth model contributes differently .

Yes, this is the idea. Each configuration distribution $\{(\eta_{ij}, \zeta_{ij}) : 1 \leq i \leq 6\}$ contributes *equally* to the distribution of the unobserved true discharge $Y_j$, but configurations with a larger logarithm of scale $\zeta_{ij}$ are less informative than configurations with a smaller $\zeta_{ij}$. We have added clarifications in the revised version of the manuscript.

6. Is Section 3.3 not assuming in some way that (conditional) uncertainty of the maximum distribution is independent of the size of the maximum? It is often the case that hydrological models are more uncertain for extremes (where the data available for calibration is more scarce): is this something that could undermine your approach?

   Uncertainty is indeed dependent on the size of the estimated discharge. Average log-discharges have a smaller $\zeta_{ij}$ than annual maxima. Our proposed model accounts for this uncertainty, and it does not undermine the approach.

7. What is the $\sigma$ which appears in equation 8 (and what value did you choose for this hyperparameter)? From the results it looks like you greatly reduce the uncertainty of the estimated maximum: could this be linked to fairly informative priors?

   The parameter $\sigma$ corresponds to the scale of the GEV distribution. The GEV distribution modeling the unobserved log-maxima has three parameters: location $\mu$, scale $\sigma$, and shape $\xi$. We did not assign any fixed values to these three parameters; they are model parameters that need to be estimated. We assigned an improper prior to $\sigma$ for Bayesian inference. Conceptually, if the true discharge annual maxima were known, the GEV parameters could be estimated using this series. In our framework, however, the true discharges are not observed, and we incorporate the associated uncertainty into the GEV parameter estimation through Eq. 8.

8. Personally I find the derivation in Section 3.5 slightly easier to follow, probably because it is closer to the traditional EIV derivation. It's OK to leave this at a later stage of the manuscript, but it is not at all clear how the two derivations are equivalent, considering the final equation has a different form (the pendices for f are different, how is this equivalent?) Also as it is the equation at line 219 is hard to parse since $\eta_i$ represents both the random variable and its realization (similarly the equation at line 215 would normally be written using the $\sim$ formulation and making clear what is a random variable and what is a realization).

   The derivation in Section 3.5 has now been adopted in response to your comment. It is now Section 3.1 and 3.2 in the revised version. Since it appears earlier in the revised manuscript, we have added more details to clarify the conditional distributions, random variables, and their realizations. We hope these changes improve clarity.

9. Section 4.3: is it surprising that MCMC samples from what you have assumed to be GEV-distributed are GEV-distributed? The real test here would be to have the measured flow values and see if the qq-plot of those values behaves like a GEV.

   It is not surprising, but it does provide some validation of the model's adequacy. The latent variables, the true unobserved discharge annual maxima, are estimated in conjunction with the GEV on one side and the distribution of the estimated discharges on the other. It is reassuring that the fitted GEV is able to model the latent series of annual maxima. This might not have been the case if the series of estimated discharge distributions were inconsistent with a single GEV distribution, which would have resulted in a poor fit.

10. Section 5.1: you provide only one value of DIC per model: did you try this across the whole river network?

    This question led us to further analyze non-stationarity, and we added Section 3.5 dedicated to non-stationary extensions. In response, we found that 100 out of the 211 river sections exhibit non-stationarity in the GEV location parameter modeling the annual discharge maxima. We now describe and discuss these new results in the revised version of the manuscript.

*These further analyses led us to realize that the DIC favored the stationary model, while the 95% posterior marginal confidence interval for $\mu_1$ did not include 0. At this point, we are unsure why the DIC might be over-penalizing non-stationary models. It may be related to the fact that the maxima are treated as missing values in the model. Perhaps another criterion, such as the Watanabe–Akaike information criterion, would be more suitable in our case. Nonetheless, the non-stationarity is now described in the manuscript using the marginal posterior distribution of the non-stationary parameters.*

11. The study would be even more convincing if it could show that the approach does indeed work as planned on simulated dataset. Since you do not use any observed data we can not really know if the new maps which have been derived are more reliable than what was there before.

*The maps are more robust because the genuine uncertainty of estimated discharges is included in the statistical model. Moreover, all the available sources of information have been also included in the proposed Bayesian hierarchical model, namely the six hydrologic model configurations.*

*In Section 4.4, we provided the 100-year return level using a simpler approach that ignores the uncertainty of estimated discharges and utilizes the available information less efficiently. While the difference between the point estimates is relatively small, the proposed approach accounts for uncertainty and provides it for the return level estimates. There is no straightforward way to estimate such uncertainty using the simpler method.*

*Comments about this point have been added in Section 4.4.*

**Minor comments**

- Line 26: a positive value "suggests" $\rightarrow$ it's a mathematical fact, so make indicates, implies, results in...

  *Done. Thank you for the pointing that out.*

- Line 29-30 "assuming climate stationarity": the definition does not require stationarity, the stationarity is needed to be able to define the return period as the quantile. See on this Volpi et al. (2015) and Salas et al. (2013).

  *Thank you for the clarification. We have added it along with your suggested references.*

- Line 114: at https://github.com/jojal5/Publications I don't see the code for this paper yet.

  *It is now available.*

- The title of Section 3.1 is not very informative

  *We changed it to True discharges as the model's missing values. We hope that it is now more informative.*

- In some equations (eg 6 and 9) the summation goes up to S, should this be a 6 (or should all the other summations go up to 6 and you should define what S is)

  *It has been fixed throughout the revised version of the manuscript. Thank you for pointing that out.*

- After Eq. 5: the function is evaluated at $y_i$, not $y$ (even if I would suggest to change Eq. 5 so that the function is evaluated at a $y$, rather than $y_i$ value for clarity)

  *It has been taken into account in the new Section 3.1 and 3.2.*

- I don't see why the mixture model in equation 9 would be a sensible idea for this.

  According to your comment, we removed this section in favor of another one that provides alternative options.

---

## Author Comment (AC2)

**EGUSPHERE-2024-2114**
**Detailed responses to Reviewer 2's comments**

Duy Anh Alexandre & Jonathan Jalbert

November 5, 2024

First, we would like to thank the Reviewer 2 who conducted a thorough review and provided relevant comments and suggestions. The modifications in the revised version of the manuscript are highlighted in blue. Below are detailed responses to all of Reviewer 2's comments.

The two reviewers suggest the use of observed discharges data to validate the results. However, it is not possible to validate the results as the reviewers suggest. The estimated discharges are obtained with a technique combining hydrologic models, interpolation and data assimilation. So, at the river sections where there is true observations, these observations are assimilated. So the real data cannot be used to validate in a strict senses the estimated discharges.

Estimated discharges at gauged sections still contain some uncertainty, primarily due to rating curve uncertainty when estimating discharges from water level measurements, especially for high discharge values. At ungauged sites, this uncertainty is greater and increases with the river distance from a gauging station. These sources of uncertainty affect the precision of return level estimates. In the manuscript, we aimed to show these impacts of uncertainty on return level estimations for both gauged and ungauged sections. To further emphasize this point in the revised version, we added Section 4.5 to analyze a gauged river section. This new analysis is twofold: first, it demonstrates the flexibility of the proposed approach to model uncertainty, and second, it shows the impact of including interpolation uncertainty in the return level estimates.

It would have been interesting to perform a cross-validation where real observations from a river section were withheld from assimilation, allowing us to compare the estimated return levels with the actual data and the estimated discharges. However, such cross-validation data were not available at the time.

That being said, the estimated discharges were already evaluated at sites with observations in Lachance-Cloutier *et al.* (2017), but the emphasis was not on annual maxima.

**Summary**

In this manuscript, the authors outline a seemingly novel method for estimating annual maximum streamflow at ungauged river sections using the results of hydrological modeling and a Bayesian inference method. The paper is generally clear and well-written, and the results are interesting, if somewhat limited in scope. I admit that I am not familiar enough with the methods to evaluate the details of the calculations. However, I find the strong emphasis in the paper on the uncertainty of estimates, as opposed to the estimated magnitudes themselves, somewhat puzzling. I feel that either the latter needs to be emphasized more, and validated against measurements if possible, or else the authors need to tell readers why it is of secondary importance compared to uncertainty reduction. This is the substance of the Principal concern detailed below.

Thank you for the review. We placed strong emphasis on describing uncertainty because our goal was to incorporate the uncertainty of the estimated discharges into the extreme value analysis. We believe that incorporating and utilizing this information results in more reliable extreme value estimates. This paper aims at describing a relatively simple method for including discharge uncertainties and combining multiple sources of information.

**Principal concern**

1. The authors emphasize the reduction in uncertainty of the annual maximum streamflow estimates at ungauged locations that results from their applied method, which propagates to estimates of extreme value distribution parameters and return levels. However, it isn't clear to me whether their method also improves their estimates of the magnitude of discharge. Isn't that equally (or more) important?

   It improves the discharge estimates by incorporating multiple sources of information: six hydrologic model configurations and the unobserved maxima from other years. The uncertainty on the true discharges is smaller than that from a single hydrological configuration due to the mild assumption of conditional independence. The method also provides an uncertainty description that would be impossible to achieve with the simpler methods discussed in Section 4.4. We added more details in Section 4.4 about this point.

2. If I understand correctly, the key results of the study are contained in Figures 2b and 5. Comparing the median values in each annmax time series, it is evident that they are systematically larger in the former series: the temporal mean appears to be roughly 1500 m3/s in Fig. 2b but < 1000 m3/s in Fig. 5. Furthermore, the confidence intervals of the posterior estimates would seem to rule out even the median values from each of the 6 simulated series shown in Fig. 2b. In other words, the differences in streamflow maxima between the two series are substantial. My question is, I believe, a very natural one: can it be established that the posterior estimates are likely to be closer to the true streamflow maxima?

   Yes this is correct. We believe that the posterior etimates are closer to the true streamflow as it efficiently combines the available information.

3. While I understand that the authors developed their method to provide information on ungauged sections of the river, I don't see why their procedure omits an evaluation step wherein one or more observed streamflow time series on gauged sections of the river are compared with simulated values at the same location, such as those shown in Figs. 2 and 5.

   The estimated discharges were already evaluated at sites with observations in Lachance-Cloutier *et al.* (2017). In sections with observations, the log-discharge uncertainty is very small, and all hydrological configurations are equivalent due to data assimilation. Therefore, comparing to data is not relevant, as the data have already been used in the discharge estimations. Nevertheless, it would have been interesting to perform a cross-validation where real observations were withheld from assimilation, allowing us to compare the estimated return levels with the actual data and the estimated discharges. However, such cross-validation data were not available at the time.

**Some specific questions I would like to see addressed are the following:**

1. Could a map be provided, similar to Figure 9, showing the locations of stream gauges where historical measurements exist? At the top of page 6, the authors refer to "... historical observed

daily values from 1961 to 2020 over more than 80 measuring stations." Were these data not available to them?

A map of the gauged river section is available at https://www.cehq.gouv.qc.ca/atlas-hydroclimatique/stations-hydrometriques/index.htm. Since we did not use the real observations directly for the reasons mentioned at the beginning of this document, we did not include this map in the manuscript. However, if the reviewers believe it is important, we would be happy to include it.

2. Would one such location be near the outlet to the Chaudière watershed, where modelled results are shown in Figures 2 and 5? If so, could the authors plot the observed annmax time series at that location, along with the simulated results, to make a comparison?

We added Section 4.5, where we analyzed the estimated discharges for a gauged section. This provides insight into the role that interpolation uncertainty plays. However, as mentioned at the beginning of this document, we cannot compare the analysis with the true observed discharges because they have already been assimilated.

3. If there are no gauge data at the outlet, then could the authors instead choose a section (or sections) of the river for modelling where measurements are available, to compare with the corresponding simulations?

Please see the response below.

4. If for some reason none of the above can be achieved, could the authors please answer the question: Why should we be impressed with the reduction in uncertainty demonstrated in Figs. 2 and 5 if we don't know that the median values in the posterior estimate are likely to be improved over the priors? It seems to me that estimating these magnitudes as accurately as possible would be the best way to ensure that, as expressed by the authors in their concluding sentence, "...the developed approach meets the practical needs of engineers responsible for updating the mapping of flood zones in Southern Québec."

We agree. Estimating the magnitude of discharges and the corresponding return levels is paramount. We showed that the uncertainty in the estimated discharges is smaller when treating them as missing values within the model, compared to estimates derived from any individual hydrological configuration. This approach results in more robust estimates. We added comments about this point in Section 4.3.